# E$^2$-Train: Training State-of-the-art CNNs with Over 80% Less Energy

**Yue Wang**$^{\diamond,*}$, **Ziyu Jiang**$^{\dagger,*}$, **Xiaohan Chen**$^{\dagger,*}$, **Pengfei Xu**$^{\diamond}$, **Yang Zhao**$^{\diamond}$,
**Yingyan Lin**$^{\diamond}$ and **Zhangyang Wang**$^{\dagger}$

$^{\dagger}$Department of Computer Science and Engineering, Texas A&M University
$^{\diamond}$Department of Electrical and Computer Engineering, Rice University
$^{\dagger}${jiangziyu, chernxh, atlaswang}@tamu.edu
$^{\diamond}${yw68, px5, zy34, yingyan.lin}@rice.edu
https://rtml.eiclab.net/?page_id=120

## Abstract

Convolutional neural networks (CNNs) have been increasingly deployed to edge devices. Hence, many efforts have been made towards efficient CNN *inference* in resource-constrained platforms. This paper attempts to explore an orthogonal direction: how to conduct more energy-efficient *training* of CNNs, so as to enable on-device training? We strive to reduce the energy cost during training, by dropping unnecessary computations, from three complementary levels: stochastic mini-batch dropping on the **data level**; selective layer update on the **model level**; and sign prediction for low-cost, low-precision back-propagation, on the **algorithm level**. Extensive simulations and ablation studies, with *real energy measurements* from an FPGA board, confirm the superiority of our proposed strategies and demonstrate remarkable energy savings for training. For example, when training ResNet-74 on CIFAR-10, we achieve aggressive energy savings of >90% and >60%, while incurring a top-1 accuracy loss of only about 2% and 1.2%, respectively. When training ResNet-110 on CIFAR-100, an over 84% training energy saving is achieved without degrading inference accuracy.

## 1 Introduction

The increasing penetration of intelligent sensors has revolutionized how Internet of Things (IoT) works. For visual data analytics, we have witnessed the record-breaking predictive performance achieved by convolutional neural networks (CNNs) [1, 2, 3]. Although such high performance CNN models are initially learned in data centers and then deployed to IoT devices, we have witnessed increasing necessity for the model to continue learning and updating itself *in situ*, such as for personalization for different users, or incremental/lifelong learning. Ideally, this learning/retraining process should take place on device. Comparing to cloud-based retraining, training locally helps avoid transferring data back and forth between data centers and IoT devices, reduce communication cost/latency, and enhance privacy.

However, training on IoT devices is non-trivial, more consuming yet much less explored than inference. IoT devices, such as smart phones and wearables, have limited computation and energy resources, that are even stringent for inference. Training CNNs consumes magnitudes higher computations than one inference. For example, training ResNet-50 for only one $224 \times 224$ image can take up to 12 GFLOPs (vs. 4GFLOPS for inference), which can easily drain a mobile phone battery when training with batch images [4]. This mismatch between the limited resources of IoT devices and the high complexity of CNNs is only getting worse because the network structures are getting more complex as they are designed to solve harder and larger-scale tasks [5].

---

$^{*}$The first three authors (Yue Wang, Ziyu Jiang, Xiaohan Chen) contributed equally.

This paper considers the most standard CNN training setting, assuming both the model structure and the dataset to be given. This "basic" training setting is not usually the realistic IoT case, but we address it as a starting point (with familiar benchmarks), and an opening door towards obtaining a toolbox that may be later extended to online/transfer learning too (see Section 5). Our goal is to **reduce the total energy cost in training**, which is complicated by a myriad of factors: from per-sample (mini-batch) complexity (both feed-forward and backward computations), to the empirical convergence rate (how many epochs it takes to converge), and more broadly, hardware/architecture factors such as data access and movements [6, 7]. Despite a handful of works on efficient, accelerated CNN training [8, 9, 10, 11, 12], they mostly focus on reducing the total **training time** in *resource-rich* settings, such as by *distributed training* in large-scale GPU clusters. In contrast, our focus is to trim down the total **energy cost** for *in-situ, resource-constrained training*. It represents an orthogonal (and less studied) direction to [8, 9, 10, 11, 12, 13, 14], although the two can certainly be combined.

To unleash the potential of more energy-efficient in-situ training, we look at the full CNN training lifecycle closely. With the goal to "squeeze out" unnecessary costs, we raise three curious questions:

- *Q1: Are all samples always required throughout training*: is it necessary to use all training samples in all epochs?
- *Q2: Are all parts of the entire model equally important during training*: does every layer or filter have to be updated every time?
- *Q3: Are precise gradients indispensable for training*: can we efficiently compute and update the model with approximate gradients?

The above three questions only represent our "first stab" ideas to explore energy-efficient training, whose full scope is much more profound. By no means do our above questions represent all possible directions. We envision that many other recipes can be blended too, such as training on lower bit precision or input resolution [15, 16]. We also recognize that energy-efficient CNN training should be jointly considered with hardware/architecture co-design [17, 18], which is beyond the current work.

Motivated by the above questions, this paper proposes a novel energy efficient CNN training framework dubbed $E^2$-**Train**. It consists of three complementary aspects of efforts to trim down unnecessary training computations and data movements, each addressing one of the above questions:

- **Data-Level: Stochastic mini-batch dropping (SMD).** We show that CNN training could be accelerated by a "frustratingly easy" strategy: randomly skipping mini-batches with 0.5 probability throughout training. This could be interpreted as data sampling with (limited) replacements, and is found to incur minimal accuracy loss (sometimes even increase).

- **Model-Level: Input-dependent selective layer update (SLU).** For each minibatch, we select a different subset of CNN layers to be updated. The input-adaptive selection is based on a low-cost gating function jointly learned during training. While similar ideas were explored in efficient inference [19], for the first time it is applied and evaluated for training.

- **Algorithm-Level: Predictive sign gradient descent (PSG).** We explore the usage of an extremely low-precision gradient descent algorithm called SignSGD, which has recently found both theoretical and experimental grounds [20]. The original algorithm still requires the full gradient computation and therefore does not save energy. We create a novel "predictive" variant, that could obtain the sign without computing the full gradient, via low-cost, bit-level prediction. Combined with mixed-precision design, it decreases computation and data-movement costs.

Besides mainly experimental explorations, we find $E^2$-Train has many interesting links to recent CNN training theories, e.g., [21, 22, 23, 24]. We evaluate $E^2$-Train in comparison with its closest state-of-the-art competitors. To measure its actual performance, $E^2$-Train is also implemented and evaluated on **an FPGA board**. The results show that the CNN model applied with $E^2$-Train consistently achieves higher training energy efficiency with marginal accuracy drops.

## 2   Related Work

**Accelerated CNN training.** A number of works have been devoted to accelerating training, in a resource-rich setting, by utilizing communication-efficient distributed optimization and larger mini-batch sizes [8, 9, 10, 11]. The latest work [12] combined distributed training with a mixed precision framework, leading to training AlexNet within 4 minutes. However, their goals and settings are distinct from ours - while the distributed training strategy can reduce time, it will actually incur more total energy overhead, and is clearly not applicable to on-device resource-constrained training.

**Low-precision training.** It is well known that CNN training can be performed under substantial lower precision [15, 14, 13], rather than using full-precision floats. Specifically, training with quantized gradients has been well studied in the distributed learning, whose main motivation is to reduce the communication cost during gradient aggregations between workers [25, 26, 27, 28, 29, 20]. A few works considered to only transmit the coordinates of large magnitudes [30, 31, 32]. Recently, the SignSGD algorithm [25, 20] even showed the feasibility of using one-bit gradients (signs) during training, without notably hampering the convergence rate or final result. However, most algorithms are optimized for distributed communication efficiency, rather than for reducing training energy costs. Many of them, including [20], need first compute full-precision gradients and then quantize them.

**Efficient CNN inference: Static and Dynamic.** Compressing CNNs and speeding up their inference have attracted major research interests in recent years. Representative methods include weight pruning, weight sharing, layer factorization, bit quantization, to just name a few [33, 34, 35, 36, 37].

While model compression presents "static" solutions for improving inference efficiency, a more interesting recent trend looks at *dynamic inference* [19, 38, 39, 40, 41] to reduce the latency, i,e, selectively executing subsets of layers in the network conditioned on each input. That sequential decision making process is usually controlled by low-cost gating or policy networks. This mechanism was also applied to improve inference energy efficiency [42, 43].

In [44], a unique bit-level prediction framework called *PredictiveNet* was presented to accelerate CNN inference at a lower level. Since CNN layer-wise activations are usually highly sparse, the authors proposed to predict those zero locations using low-cost bit predictors, thereby bypassing a large fraction of energy-dominant convolutions without modifying the CNN structure.

Energy-efficient training is different from and more complicated than its inference counterpart. However, many insights gained from the latter can be lent to the former. For example, the recent work [45] showed that performing active channel pruning during training can accelerate the empirical convergence. Our proposed model-level SLU is inspired by [19]. The algorithm-level PSG also inherits the idea of bit-level low-cost prediction from [44].

## 3 The Proposed Framework

### 3.1 Data-Level: Stochastic mini-batch dropping (SMD)

We first adopt a straightforward, seemingly naive, yet surprisingly effective stochastic mini-batch dropping (SMD) strategy (see Fig. 1), to aggressively reduce the training cost by letting it see less mini-batches. At each epoch, SMD simply skips every mini-batch with a default probability of $0.5$. All other training protocols, such as learning rate schedule, remain unchanged. Compared to the normal training, SMD can directly half the training cost, if both were trained with the same number of epochs. Yet amazingly, we observe in our experiments that SMD usually leads to negligible accuracy decrease, sometimes even increase (see Sec. 4). Why? We discuss possible explanations below.

SMD can be interpreted as sampling *with limited replacement*. To understand this, think of combing two consecutive SMD-enforced epochs into one, then it has the same number of mini-batches as one full epoch; but within it each training sample now has 0.25, 0.5, and 0.25 probability, to be sampled 2, 1, and 0 times, respectively. The conventional wisdom is that for stochastic gradient descent (SGD), in each epoch, the mini-batches are sampled i.i.d. from data *without replacement* (i.e., each sample occurs exactly once per epoch) [46, 47, 48, 49, 50]. However, [21] proved that sampling mini-batches *with replacement* has a large variance than sampling without replacement, and consequently SGD may have better regularization properties.

Alternatively, SMD could also be viewed as a special data augmentation way that injects more sampling noise to perturb training distribution every epoch. Past works [51, 52, 53] have shown that specific kinds of random noise aid convergence through escaping from saddle points or less generalizable minima. The structured sampling noise caused by SMD might aid the exploration.

Besides, [22, 54, 55] also showed that an importance sampling scheme that focuses on training more with "informative" examples leads to faster convergence under resource budgets. They implied that the mini-batch dropping can be selective based on certain information criterion instead of stochastic. We use SMD because it has zero overhead, but more effective dropping options might be available if low-cost indicators of mini-batch importance can be identified: we leave this as future work.

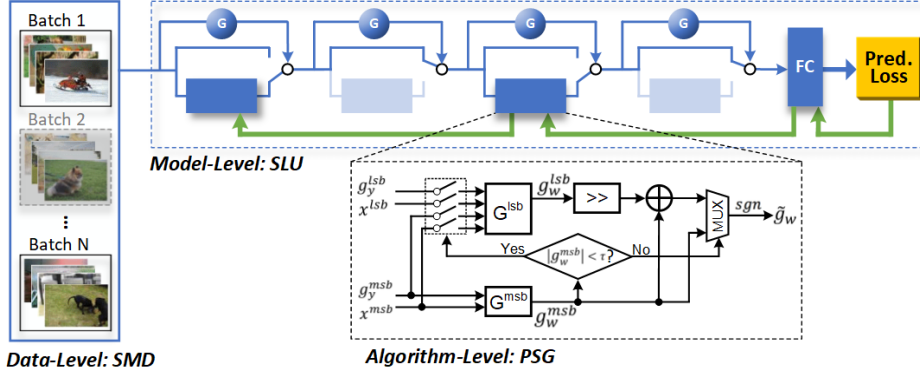

**Figure 1:** Illustration of proposed framework. SLU: each blue circle $G$ indicates an RNN gate and each blue square under $G$ indicates one block of layers in the base model. Green arrows denote the backward propagation. To reduce the training cost, the RNN gates generate strategies to select which layers to train for each input. In this specific example, the second and fourth blocks are "skipped" for **both feedforward and backward computations**. Only the first and third blocks are updated. SMD and PSG: details are described in the main text.

## 3.2 Model-Level: Input-dependent selective layer update (SLU)

[19] proposed to dynamically skip a subset of layers for different inputs, in order to adaptively accelerate the feed-forward inference. However, [19] called for a post process after supervised training, i.e., to refine the dynamic skipping policy via reinforcement learning, thus causing undesired extra training overhead. We propose to extend the idea of dynamic inference to the training stage, i.e., dynamically skipping a subset of layers during **both feed-forward and back-propagation**. Crucially, we show that by adding an *auxiliary regularization*, such dynamic skipping can be learned from scratch and obtain satisfactory performance: **no post refinement nor extra training iterations is required**. That is critical for dynamic layer skipping to be useful for energy-efficient training: we term this extended scheme as input-dependent selective layer update (SLU).

As depicted in Fig. 1, given a base CNN to be trained, we follow [19] to add a light-weight RNN gating network per layer block. Each gate takes the same input as its corresponding layer, and outputs soft-gating outputs between [0,1] for the layer, which are then used as the skipping probability, in which the higher the value is, more probably that layer will be selected. Therefore, each layer will be adaptively selected or skipped, depending on the inputs. We will only select the layers activated by gates. Those RNN gates cost less than 0.04% feed-forward FLOPs than the base models; hence their energy overheads are negligible. More details can be found in the supplementary.

[19] first trained the gates in a supervised way together with the base model. Observing that such learned routing policies were often not sufficiently efficient, they used reinforcement learning post-processing to learn more aggressive skipping afterwards. While this is fine for the end goal of dynamic inference, we hope to get rid of the post-processing overhead. We incorporate the computational complexity regularization into the objective function to overcome this hurdle, defined as

$$\min_{W,G} \ L(W,G) \ + \alpha C(W,G) \tag{1}$$

Here, $\alpha$ is a weighting coefficient of the computational complexity regularization. $W$ and $G$ denote the parameters of the base model and the gating network, respectively. Also, $L(W,G)$ denotes the prediction loss, and $C(W,G)$ is calculated by accumulating the computational cost (FLOPs) of the layers that are selected. The regularization explicitly encourages to learn more "parismous" selections throughout the training. We find that such SLU-regularized training leads to almost the same number of epochs to converge compared to standard training, i.e., SLU does not sacrifice empirical convergence speed. As a side effect, SLU will naturally yield CNNs with dynamic inference capability. Though not the focus of this paper, we find the CNN trained with SLU reaches comparable accuracy-efficiency trade-off over one trained with the approach in [19].

The practice of SLU seems to align with several recent theories on CNN training. In [56], the authors suggested that "not all layers are created equal" for training. Specifically, some layers are critical to be intensively updated for improving final predictions, while others are insensitive along training. There exist "non-critical" layers that barely change their weights throughout training: even resetting those layers in a trained model to their initial value has few negative consequences. The more recent work [24] further confirmed the phenomenon, though how to identify those non-critical model parts at the

early training stage remains unclear. [57, 58] also observed different samples might activate different sub-models. Those inspiring theories, combined with the dynamic inference practice, motivate us to propose SLU for more efficient training.

## 3.3 Algorithm-Level: Predictive sign gradient descent (PSG)

It is well recognized that low-precision fixed-point implementation is a very effective knob for achieving energy efficient CNNs, because both CNNs' computational and data movement costs are approximately a quadratic function of their employed precision. For example, a state-of-the-art design [59] shows that adopting 8-bit precision for a multiplication, adder, and data movement can reduce the energy cost by 95%, 97%, and 75%, respectively, as compared to that of a 32-bit floating point design when evaluated in a commercial 45nm CMOS technology.

The successful adoption of extremely low-precision (binary) gradients in SignSGD [20] is appealing, as it might lead reducing both weight update computation and data movements. However, directly applying the original SignSGD algorithm for training will not save energy, because it actually computes the full-precision gradient first before taking the signs. We propose a novel predictive sign gradient descent (PSG) algorithm, which predicts the sign of gradients using low-cost bit-level predictors, therefore completely bypassing the costly full-gradient computation.

We next introduce how the gradients of weights are updated in PSG. Assume the following notations: the full precision and **m**ost **s**ignificant **b**its (the latter, **MSB** part, is adopted by PSG's low-cost predictors) of the input $x$ and the gradient of the output $g_y$ are denoted as $(B_x, B_g)$ and $(B_x^{\mathrm{msb}}, B_g^{\mathrm{msb}})$, respectively, where the corresponding input and the gradient of the output for PSG's predictors are denoted as $x^{\mathrm{msb}}$ and $g_y^{\mathrm{msb}}$, respectively. As such, the quantization noise for the input and the gradient of the output are $q_x = x - x^{\mathrm{msb}}$ and $q_{g_y} = g_y - g_y^{\mathrm{msb}}$, respectively. Similarly, after back-propagation, we denote the full-precision and low-precision (i.e., taking the most significant bits (MSBs)) gradient of the weight as $g_w$ and $g_w^{\mathrm{msb}}$, respectively, the latter of which is computed using $x^{\mathrm{msb}}$ and $g_y^{\mathrm{msb}}$. Then, with an empirically pre-selected threshold $\tau$, PSG updates the $i$-th weight gradient as follows:

$$\tilde{g}_w[i] = \begin{cases} sgn(g_w^{\mathrm{msb}}[i]) & , |g_w^{\mathrm{msb}}[i]| \geq \tau \\ sgn(g_w[i]) & , \text{otherwise} \end{cases} \tag{2}$$

Note that in hardware implementation, the computation to obtain $g_w^{\mathrm{msb}}$ is embedded within that of $g_w$. Therefore, the PSG's predictors do not incur energy overhead.

**PSG for energy-efficient training.** Recent work [15] has shown that most of the training process is robust to reduced precision (e.g., 8 bits instead of 32 bits), except for the weight gradient calculations and updates. Taking their learning, we similarly adopt a higher precision for the gradients than the inputs and weights, i.e., $B_{g_y} > B_x = B_w$. Specifically, when training with PSG, we first compute the predictors using $B_x^{\mathrm{msb}}$ (e.g., $B_x^{\mathrm{msb}} = 4$) and $B_{g_y}^{\mathrm{msb}}$ (e.g., $B_{g_y}^{\mathrm{msb}} = 10$), and then update the weights' gradients following Eq. (2). The further energy savings of training with PSG over the fixed-point training [15] are resulted from the fact that the predictors computed using $x^{\mathrm{msb}}$ and $g_y^{\mathrm{msb}}$ require exponentially less computational and data movement energy.

**Prediction guarantee of PSG.** We analyze the probability of PSG's prediction failure to discuss its performance guarantee. Specifically, if denoting the sign prediction failure produced by Eq. (2) as $H$, it can be proved that this probability is upbounded as follows,

$$P(H) \leq \Delta_x^2 E_1 + \Delta_{g_y}^2 E_2, \tag{3}$$

where $\Delta_x = 2^{-(B_x^{\mathrm{msb}} - 1)}$ and $\Delta_{g_y} = 2^{-(B_{g_y}^{\mathrm{msb}} - 1)}$ are the quantization noise step sizes of $x^{\mathrm{msb}}$ and $g_y^{\mathrm{msb}}$, respectively. $E_1$ and $E_2$ are given in the Appendix along with the proof of Eq. (3). Eq. (3) shows that the prediction failure probability of PSG is upbounded by a term that degrades exponentially with the precision assigned to the predictors, indicating that this failure probability can be very small if the predictors are designed properly.

**Adaptive threshold.** Training with PSG might lead to sign flips in the weight gradients as compared to that of the floating point one, which only occurs when the latter has a small magnitude and thus the quantization noise of the predictors causes the sign flips. Therefore, it is important to properly select a threshold (e.g., $\tau$ in Eq.(2)) that can optimally balance this sign flip probability and the achieved energy savings. We adopt an adaptive threshold selection strategy because the dynamic range of gradients differ significantly from layers to layers: instead of using a fixed number, we will tune a ratio $\beta \in (0, 1)$ which yields the adaptive threshold as $\tilde{\tau} = \beta \max_i \{g_w^{\mathrm{msb}}[i]\}$.

# 4 Experiments

## 4.1 Experiment setup

Datasets: We evaluate our proposed techniques on two datasets: CIFAR-10 and CIFAR-100. Common data augmentation methods (e.g., mirroring/shifting) are adopted, and data are normalized as in [60]. Models: Three popular backbones, ResNet-74, ResNet-110 [61], and MobileNetV2 [62], are considered. For evaluating each of the three proposed techniques (i.e., SMD, SLU, and PSG), we consider various experimental settings using ResNet-74 and CIFAR-10 dataset for ablation study, as described in Sections 4.2-4.5. ResNet-110 and MobileNetV2 results are reported in Section 4.6. Top-1 accuracies are measured for CIFAR-10, and both top-1 and top-5 accuracies for CIFAR-100. Training settings: We adopt the training settings in [61] for the baseline default configurations. Specifically, we use an SGD with a momentum of 0.9 and a weight decaying factor of 0.0001, and the initialization introduced in [63]. Models are trained for 64k iterations. For experiments where PSG is used, the initial learning rate is adjusted to 0.03 as SignSGD[20] suggested small learning rates to benefit convergence. For others, the learning rate is initially set to be 0.1 and then decayed by 10 at the 32k and 48k iterations, respectively. We also employed the stochastic weight averaging (SWA) technique [64] when PSG is adopted, that was found to notably stabilize training.

Real energy measurements using FPGA: As the energy cost of CNN inference/training consists of both computational and data movement costs, the latter of which is often dominant but can not captured by the commonly used metrics, such as the number of FLOPs [6], we evaluate the proposed techniques against the baselines in terms of accuracy and real measured energy consumption. Specifically, unless otherwise specified, all the energy or energy savings are obtained **through real measurements** by training the corresponding models and datasets in a state-of-the-art FPGA [65], which is a digilent ZedBoard Zynq-7000 ARM/FPGA SoC Development Board. Fig. 2 shows our FPGA measurement setup, in which the FPGA board is connected to a laptop through a serial port and a power meter. In particular, the training settings are downloaded from the laptop to the FPGA board, and the real-measured energy consumption is obtained via the power meter for the whole training process and then sent back to the laptop. **All energy results are measured from FPGA.**

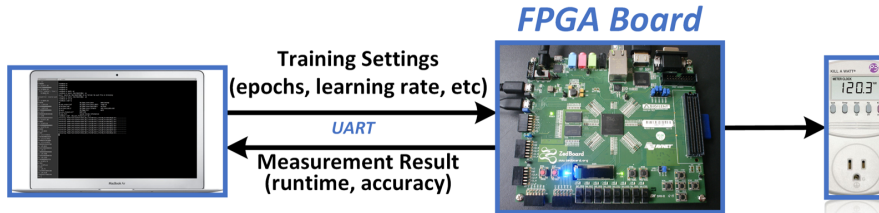

**Figure 2:** The energy measurement setup with (from left to right) a MAC Air latptop, a Xilinx FPGA board [65], and a power meter.

## 4.2 Evaluating stochastic mini-batch dropping

We first validate the energy saving achieved by SMD against a few "off-the-shelf" options: (1) can we train with the standard algorithm, using less iterations and otherwise the same training protocol? (2) can we train with the standard algorithm, using less iterations but properly increased learning rates? Two set of carefully-designed experiments are presented below for addressing them.

**Training with SMD vs. standard mini-batch (SMB):** We first evaluate SMD over the standard mini-batch (SMB) training, which uses all (vs. 50% in SMD) mini-batch samples. As shown in Fig. 3a when the energy ratio is 1 (i.e., training with SMB + 64k iterations vs. SMD + 128k iterations), the proposed SMD technique is able to boost the inference accuracy by 0.39% over the standard way.

We next "naively" suppress the energy cost of SMB, by reducing training iterations. Specifically, we reduce the SMB training iterations to be $\{\frac{6}{12}, \frac{7}{12}, \frac{8}{12}, \frac{9}{12}, \frac{10}{12}, \frac{11}{12}, 1\}$ of the original one. Note the learning rate schedule (e.g., when to reduce learning rates) will be scaled proportionally with the total iteration number too. For comparison, we conduct experiments of training with SMD when the number of equivalent training iterations are the same as those of the SMB cases. Fig. 3a shows that training with SMD consistently achieves a higher inference accuracy than SMB with the margin ranging from 0.39% to 0.86%. Furthermore, we can see that training with SMD reduces the training energy cost by 0.33 while boosting the inference accuracy by 0.2% (see the cases of SMD under the energy ratio of 0.67 vs. SMB under the energy ratio of 1, respectively, in Fig. 3a), as compared to SMB. We adopt SMD under this energy ratio of 0.67 in all the remaining experiments.

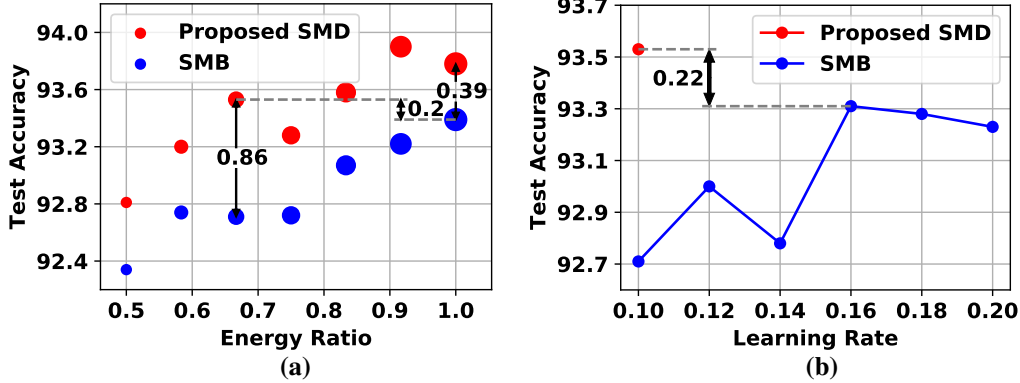

**Figure 3:** The top-1 accuracy of CIFAR-10 testing using the Resnet-74 model when: (a) training with SMD versus the standard mini-batch (SMB) method, with the two's training energy ratio ranging from 0.5 to 1. The size of markers is drawn proportionally with the measured training energy cost; and (b) training with SMD, and SMB with different increased learning rates – all under the same training energy budget.

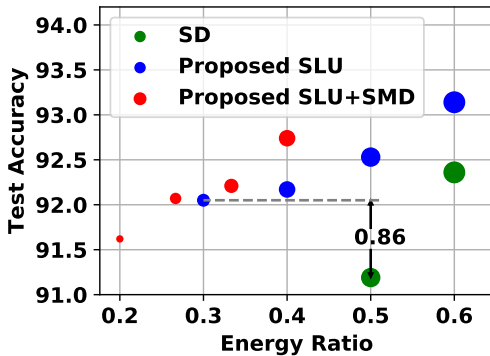

**Figure 4:** Inference accuracy vs. energy ratios, where the energy ratios are obtained by normalizing the corresponding energy over that of the original one (SMB + 64k iterations).

**Table 1:** The accuracy of SMD on other datasets and backbones (energy ratio 0.67).

| Dataset | Backbone | Accuracy | |
|---|---|---|---|
| | | SMB | SMD |
| CIFAR-10 | ResNet-110 | 92.75% | **93.05%** |
| CIFAR-100 | ResNet-74 | 71.11% | **71.37%** |

We repeated training ResNet-74 on CIFAR-10 using SMD for 10 times with different random initializations. The accuracy standard deviation is only 0.132%, showing high stability. We also conducted more experiments with different backbones and datasets . As shown in Tab. 1, SMD is consistently better than SMB.

**Training with SMD vs. SMB + increased learning rates:** We further compare with SMB with tuned/larger learning rates, conjecturing that it would accelerate convergence by possibly reducing the needed training epochs. Results are summarized in Fig. 3b. Specifically, when the number of iterations are reduced by $\frac{1}{3}$, we do a grid search of learning rates, with a step size from 0.02 between [0.1,0.2]. All compared methods are set with the same training energy budget. Fig. 3b demonstrates that while increasing learning rates seem to improve SMB's energy efficiency over sticking to the original protocol, our proposed SMD still maintains a clear advantage of at least 0.22%.

### 4.3 Evaluating selective layer update

Our current SLU experiments are based on CNNs with residual connections, partially because they dominate in SOTA CNNs. We will extend SLU to other model structures in future work. We evaluate the proposed SLU by comparing it with stochastic depth (SD) [66], a technique originally developed for training very deep networks effectively, by updating only a random subset of layers at each mini-batch. It could be viewed as a "random" version of SLU (which uses learned layer selection). We follow all suggested settings in [66]. For a fair comparison, we adjust the hyper-parameter $p_L$ [66], so that SD dropping ratio is always the same as SLU.

From Fig. 4, training with SLU consistently achieves higher inference accuracies than SD when their training energy costs are the same. It is further encouraging to observe that training with SLU could even achieve higher accuracy sometimes in addition to saving energy. For example, comparing the cases when training with SLU + an energy ratio of 0.3 (i.e., 70% energy saving) and that of SD + an energy ratio of 0.5, the proposed SLU technique is able to reduce the training energy cost by 20% while boosting the inference accuracy by 0.86%. These results endorses the usage of data-driven gates instead of random dropping, in the context of energy-efficient training. Training with SLU + SMD combined further boosts the accuracy while reducing the energy cost. Furthermore, 20

trials of SLU experiments to ResNet38 on CIFAR-10 conclude that, with 95% confidence level, the confidence interval for the mean of the top-1 accuracy and the energy saving are [92.47%, 92.58%] (baseline:92.50%) and [39.55%, 40.52%], respectively, verifying SLU's trustworthy effectiveness.

## 4.4 Evaluating predictive sign gradient descent

We evaluate PSG against two alternatives: (1) 8-bit fixed point training proposed in [15]; and (2) the original SignSGD [20]. For all experiments in Sections 4.4 and 4.5, we adopt 8-bit precision for the activations/weights and 16-bit for the gradients. The corresponding precision of the predictors are 4-bit and 10-bit, respectively. We use an adaptive threshold (see Section 3.3) of $\beta=0.05$. More experiment details are in Appendix.

**Table 2:** Comparing the inference accuracy and achieved energy savings (over the 32-bit floating point training) when training with SGD, 8-bit fixed point [15], SignSGD, and PSG using Resnet-74 and CIFAR-10.

| Method | 32-bit SGD | 8-bit[15] | SignSGD[20] | PSG |
|---|---|---|---|---|
| Accuracy | 93.52% | 93.24% | 92.54% | 92.59% |
| Energy savings | - | 38.62% | - | 63.28% |

**Table 3:** The inference accuracy and energy savings (over the 32-bit floating point training) of the proposed $E^2$-Train under different (averaged) SLU skipping ratios and adaptive thresholds (i.e., $\beta$ in Section 3.3) when using Resnet-74 and CIFAR-10.

| Skipping | 20% | 40% | 60% |
|---|---|---|---|
| Accuracy ($\beta=0.05$) | 92.12% | 91.84% | 91.36% |
| Accuracy ($\beta=0.1$) | 92.15% | 91.72% | 90.94% |
| Computational savings | 80.27% | 85.20% | 90.13% |
| Energy savings | 84.64% | 88.72% | 92.81% |

As shown in Table 2, while the 8-bit fixed point training in [15] saves about 39% training energy (going from 32-bit to 8-bit in general leads to about 80% energy saving, which is compromised by their employed 32-bit gradients in this case) with a marginal accuracy loss of 0.28% as compared to the 32-bit SGD, the proposed PSG almost doubles the training energy savings (63% vs. 39% for [15]) with a negligible accuracy loss of 0.65% (93.24% vs. 92.59% for [15]). Interestingly, PSG slightly boosts the inference accuracy by 0.05% while saving 63% energy, i.e., **3×** better training energy efficiency with a slightly better inference accuracy, compared to SignSGD [20]. Besides, as we observed, the ratio of using $g_w^{\mathrm{msb}}$ for sign prediction typically remains at least 60% throughout the training process, given adaptive threshold $\beta = 0.05$.

## 4.5 Evaluating $E^2$-Train: SMD + SLU + PSG

We now evaluate the proposed $E^2$-Train framework, which combines the SMD, SLU, and PSG techniques. As shown in Table 3, we can see that $E^2$-Train: (1) indeed can further boost the performance as compared to training with SMD+SLU (e.g., $E^2$-Train achieves a higher accuracy of 0.5% (92.1% vs. 91.6% (see Fig.4 at the energy ratio of 0.2) of training with SMD+SLU, when achieving 80% energy savings); and (2) can achieve an extremely aggressive energy savings of > 90% and > 60%, while incurring an accuracy loss of only about 2.0% and 1.2%, respectively, as compared to that of the 32-bit floating point SGD (see Table 2), i.e., up to **9× better training energy efficiency with small accuracy loss**.

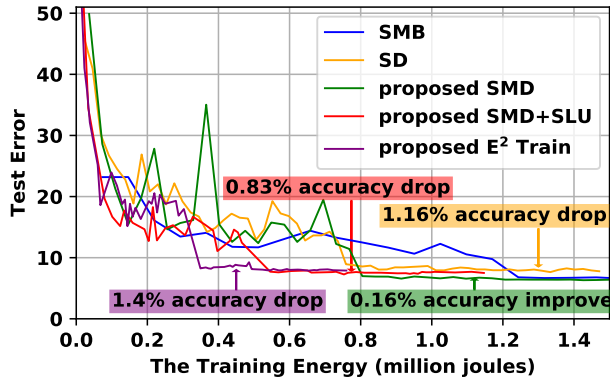

**Figure 5:** Inference accuracy vs. energy costs, at different stages of training, (i.e., the empirical convergence curves), when training with SMB, SD, merely SLU, SLU + SMD, and $E^2$-Train on CIFAR-10 with ResNet-74.

**Impact on empirical convergence speed.** We plot the training convergence curves of different methods in Fig. 5, with the x-axis represented in the alternative form of training energy costs (up to current iteration). We observe that $E^2$-Train does not slow down the empirical convergence. In fact, it even makes the training loss decrease faster in the early stage.

**Experiments on adapting a pre-trained model.** We perform a proof-of-concept experiment for CNN fine-tuning by splitting CIFAR-10 training set into half, where each class was i.i.d. split evenly. We first pre-train ResNet-74 on the first half, then fine-tune it on the second half. During fine-tuning, we compare two energy-efficient options: (1) fine-tuning only the last FC layer using

standard training; (2) fine-tuning all layers using $E^2$-Train. With all hyperparameters being tuned to best efforts, the two fine-tuning methods improve over the pre-trained model top-1 accuracy by [0.30%, 1.37%] respectively, while (2) saves 61.58% more energy (FPGA-measured) than (1). That shows that $E^2$-Train is the preferred option: higher accuracy and more energy savings

Table 4 evaluates $E^2$-Train and its ablation baselines on various models and more datasets. The conclusions are aligned with the ResNet-74 cases. Remarkably, on CIFAR-10 with ResNet-110, $E^2$-Train saves over 83% energy with only 0.56% accuracy loss. When saving over 91% (i.e., more than $10\times$), the accuracy drop is still less than 2%. On CIFAR-100 with ResNet-110, $E^2$-Train can even surpass baseline on both top-1 and top5 accuracy while saving over 84% energy. More notably, $E^2$-Train is also effective for even compact networks: it saves about 90% energy cost while achieving a comparable accuracy, when adopted for training MobileNetV2.

**Table 4:** Experiment results with ResNet-110 and MobileNetV2 on CIFAR-10/CIFAR-100.

| Dataset | Method | Backbone | Computational Savings | Energy Savings | Accuracy (top-1) | Accuracy (top-5) |
|---|---|---|---|---|---|---|
| CIFAR-10 | SMB (original) SD[66] | ResNet-110 | - 50% | - 46.03% | 93.57% 91.51% | - - |
| | SMB (original) | MobileNetV2[67] | - | - | 92.47% | - |
| | $E^2$-Train (SMD+SLU+PSG) | ResNet-110 | 80.27% 85.20% 90.13% | 83.40% 87.42% 91.34% | 93.01% 91.74% 91.68% | - - - |
| | | MobileNetV2[67] | 75.34% | 88.73% | 92.06% | - |
| CIFAR-100 | SMB (original) SD[66] | ResNet-110 | - 50% | - 48.34% | 71.60% 70.40% | 91.50% 92.58% |
| | SMB (original) | MobileNetV2[67] | - | - | 71.91% | - |
| | $E^2$-Train (SMD+SLU+PSG) | ResNet-110 | 80.27% 85.20% 90.13% | 84.17% 88.72% 92.90 % | 71.63% 68.61% 67.94% | 91.72% 89.84% 89.06% |
| | | MobileNetV2[67] | 75.34% | 88.17% | 71.61% | - |

## 5 Discussion of Limitations and Future Work

We propose the $E^2$-Train framework to achieve energy-efficient CNN training in resource-constrained settings. Three complementary aspects of efforts to trim down training costs - from data, model and algorithm levels, respectively, are carefully designed, justified, and integrated. Experiments on both simulation and real FPGA demonstrate the promise of $E^2$-Train. Despite the preliminary success, we are aware of several limitations of $E^2$-Train, which also points us to the future road map. For example, $E^2$-Train is currently designed and evaluated for standard off-line CNN training, with all training data presented in batch, for simplicity. This is not scalable for many real-world IoT scenarios, where new training data arrives sequentially in a stream form, with limited or no data buffer/storage leading to the open challenge of "on-the-fly" CNN training [68]. In this case, while both SLU and PSG are still applicable, SMD needs to be modified, e.g., by one-pass active selection of stream-in data samples. Besides, SLU is not yet straightforward to be extended to plain CNNs without residual connections. We expect finer-grained selective model updates, such as online channel pruning [45], to be useful alternatives here. We also plan to optimize $E^2$-Train for continuous adaptation or lifelong learning.

**Acknowledgments**

The work is in part supported by the NSF RTML grant (1937592, 1937588). The authors would like to thank all anonymous reviewers for their tremendously useful comments to help improve our work.

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
