[Supplementary Material]



# E$^2$-Train: Energy-Efficient Deep Network Training with Data-, Model-, and Algorithm-Level Saving (Supplementary Material)

## A  PSG Prediction Error Rate Bound Analysis

In this section we analyze the probability of a sign prediction failure bound (3) in PSG (2).

**Weight gradient calculation during back propagation.** Consider we have a convolutional layer with weight $w$ and no bias (as is the usual case for modern deep CNNs), its input is $x$ and the output is $y$. During one pass of back propagation, the gradient propagated by its succeeding layer is $g_y$. We compute the gradient of the weight as $g_w = g_w(x, g_y)$. Considering only one entry in $g_w$, it can be represented by the sum of a series inner product of the corresponding locations in $x$ and $g_y$. For simplicity and with a little abuse of notations, the one entry the gradient can be represented as:

$$g_w = \sum_{n=1}^{N} x_n^T g_{y,n}, \tag{4}$$

where $n$ iterate over the mini-batch and $N$ is the mini-batch size. The MSB parts used to predict the gradient signs are denoted as $x^{\text{msb}}$ and $g_y^{\text{msb}}$, with precision $B_x^{\text{msb}}$ and $B_{g_y}^{\text{msb}}$. The corresponding quantization noise terms are $q_x$ and $q_{g_y}$. The gradient calculated using (4) with $x^{\text{msb}}$ and $g_y^{\text{msb}}$ is denoted as $g_w^{\text{msb}}$. Then the gradient error, denoted as $q_w$, can be approximated with

$$q_w = \sum_{n=1}^{N} \left( x_n^T q_{g_y n} + q_{x,n}^T g_{y,n} \right). \tag{5}$$

Here the second order noise term is neglected because it is small.

**Sign prediction error probability bound.** Denote the the sign prediction failure, given a t event as $H$, which has three subcases $H_0, H_p, H_n$ as shown in Table 4:

| Event | Condition |
|:---:|:---:|
| $H_0$ | $g_w = 0, \|g_w^{\text{msb}}\| > \tau$ |
| $H_p$ | $g_w > 0, g_w^{\text{msb}} < -\tau$ |
| $H_n$ | $g_w < 0, g_w^{\text{msb}} > \tau$ |

**Table 4:** Three cases when a sign prediction error happens in PSG.

**Consider Case $H_0$:**

$$
\begin{aligned}
P(H_0) &= P(g_w = 0, |g_w^{\text{msb}}|) > \tau) \\
&= P(g_w = 0)P(|g_w^{\text{msb}}| > \tau | g_w = 0) \\
&= P(g_w = 0)P(|q_w| > \tau | g_w = 0) \\
&= P(g_w = 0) \int f_{\mathbf{X}|g_w=0}(\mathbf{x}) P\left(|q_w| > \tau | g_w = 0, \mathbf{X} = \mathbf{x}\right) d\mathbf{x} \\
&\leq \frac{P(g_w = 0)}{\tau^2} \int f_{\mathbf{X}|g_w=0}(\mathbf{x}) \sigma^2(q_w) d\mathbf{x},
\end{aligned}
\tag{6}
$$

where $f_{\mathbf{X}|g_w=0}(\mathbf{x})$ is the conditional distribution of $\mathbf{X}$ given $g_w = 0$, and $\sigma^2(q_w)$ is the variance of $q_w$. The inequality comes from Chebychev's inequality and the fact that $q_w$ is symmetrically

distributed. Plug $\sigma^2(q_w) = \frac{1}{12}\sum_{n=1}^{N}\left(\Delta_x^2\|g_{y,n}\|^2 + \Delta_{g_y}^2\|x_n\|^2\right)$ into (6), we have:

$$P(H_0) \leq \frac{P(g_w = 0)}{12\tau^2}\int f_{\mathbf{X}|g_w=0}(\mathbf{x})\sum_{n=1}^{N}\left(\Delta_x^2\|g_{y,n}\|^2 + \Delta_{g_y}^2\|x_n\|^2\right)d\mathbf{x}$$

$$= \frac{P(g_w = 0)}{12\tau^2}E\left[\sum_{n=1}^{N}\left(\Delta_x^2\|g_{y,n}\|^2 + \Delta_{g_y}^2\|x_n\|^2\right)\Bigg|g_w = 0\right]$$

$$= \frac{1}{12\tau^2}E\left[\sum_{n=1}^{N}\left(\Delta_x^2\|g_{y,n}\|^2 + \Delta_{g_y}^2\|x_n\|^2\right)\cdot\mathbb{1}_{g_w=0}\right]$$

$$= \frac{\Delta_x^2}{12\tau^2}\sum_{n=1}^{N}E\left[\|g_{y,n}\|^2\cdot\mathbb{1}_{g_w=0}\right] + \frac{\Delta_{g_y}^2}{12\tau^2}\sum_{n=1}^{N}E\left[\|x_n\|^2\cdot\mathbb{1}_{g_w=0}\right]. \qquad (7)$$

**Consider Case $H_p$ and $H_n$**: following similar derivations, we can have:

$$P(H_p) \leq \frac{\Delta_x^2}{24}\sum_{n=1}^{N}E\left[\frac{\|g_{y,n}\|^2\cdot\mathbb{1}_{g_w>0}}{\left[\sum_{n=1}^{N}\left(x_n^T q_{g_y n} + q_{x,n}^T g_{y,n}\right) + \tau\right]^2}\right]$$

$$+ \frac{\Delta_{g_y}^2}{24}\sum_{n=1}^{N}E\left[\frac{\|x_n\|^2\cdot\mathbb{1}_{g_w>0}}{\left[\sum_{n=1}^{N}\left(x_n^T q_{g_y n} + q_{x,n}^T g_{y,n}\right) + \tau\right]^2}\right], \qquad (8)$$

$$P(H_n) \leq \frac{\Delta_x^2}{24}\sum_{n=1}^{N}E\left[\frac{\|g_{y,n}\|^2\cdot\mathbb{1}_{g_w<0}}{\left[\sum_{n=1}^{N}\left(x_n^T q_{g_y n} + q_{x,n}^T g_{y,n}\right) + \tau\right]^2}\right]$$

$$+ \frac{\Delta_{g_y}^2}{24}\sum_{n=1}^{N}E\left[\frac{\|x_n\|^2\cdot\mathbb{1}_{g_w<0}}{\left[\sum_{n=1}^{N}\left(x_n^T q_{g_y n} + q_{x,n}^T g_{y,n}\right) + \tau\right]^2}\right]. \qquad (9)$$

Combining (7-9), we get the probability bound of a sign prediction failure

$$P(H) = P(H_0) + P(H_p) + P(H_n) \leq \Delta_x^2 E_1 + \Delta_{g_y}^2 E_2,$$

where $E_1$ and $E_2$ are defined as:

$$E_1 \leq \frac{1}{12\tau^2}\sum_{n=1}^{N}E\left[\|g_{y,n}\|^2\cdot\mathbb{1}_{g_w=0}\right] + \frac{1}{24}\sum_{n=1}^{N}E\left[\frac{\|g_{y,n}\|^2\cdot\mathbb{1}_{g_w\neq 0}}{\left[\sum_{n=1}^{N}\left(x_n^T q_{g_y n} + q_{x,n}^T g_{y,n}\right) + \tau\right]^2}\right],$$

$$E_2 \leq \frac{1}{12\tau^2}\sum_{n=1}^{N}E\left[\|g_{y,n}\|^2\cdot\mathbb{1}_{g_w=0}\right] + \frac{1}{24}\sum_{n=1}^{N}E\left[\frac{\|x_n\|^2\cdot\mathbb{1}_{g_w\neq 0}}{\left[\sum_{n=1}^{N}\left(x_n^T q_{g_y n} + q_{x,n}^T g_{y,n}\right) + \tau\right]^2}\right].$$

**Discussion of the data range.** In (3) the data range is assumed to be $[-1, 1]$. When the data range changes, however, the bound will not change because it is equivalent with scaling the numerators and denominators in the derivations above, which corresponds to the adaptive threshold scheme we introduce in Section 3.3.

## B  Experiment Settings for PSG in Section 4.4

Instead of using the default training settings described in Section 4.1, we use a learning rate of 0.03 and a weight decay of 0.0005 for SignSGD [15] and PSG in Section 4.4, which we found optimal

for most cases when SignSGD was involved (PSG also uses SignSGD because it predicts the sign to replace weight gradients). During the experiments, we found it a little bit tricky to find a suitable learning rate. Because both of SignSGD and PSG use the sign of the gradients to update weights, they demand smaller learning rate especially when the performance improves and gradients approach to near zero. The above setting is consistent to the observations in [15] that the learning rate for SignSGD should be appropriately smaller than that for the baseline algorithm.

## C  SLU Implementation Details

In our implementation, we adopt the recurrent gates (RNNGates) as in [14]. It is composed of a global average pooling followed by a linear projection that reduces the features to a 10-dimensional vector as depicted in 6. A Long Short Term Memory (LSTM) [61] network that contains a single layer of dimension 10 is applied to generate a binary scalar. As mentioned in [14], this RNN gating networks design incurs a negligible overhead compared to its feed-forward counterpart (0.04% vs. 12.5% of the computation of

**Figure 6:** Gating networks in SLU are RNNs that share weights (RNNGates). The RNNGates incurs a negligible overhead.

the residual blocks when the baseline architecture is a ResNet). In order to further reduce the energy cost due to loading parameters into the memory, all RNNGates in the SLU share the same weights.

**Training with SLU + SMD:** We further evaluate the performance of combing the SLU and SMD techniques. As shown in Fig. 5, training with SLU + SMD consistently boost the inference accuracy further while reducing the training energy cost. For example, compared to the SD baseline, SLU + SMD can improve the inference accuracy by **0.43%**, while costing **60%** lower energy.