[Reviews · NeurIPS 2019]

Reviewer 1



I read through the rebuttal and peer reviewers' comments. I think the authors did an excellent job in addressing all raised concerns. Their responses are informative and convincing (with lots of new results as requested). In particular, the authors were able to further eliminate the accuracy drop (by digging out and incorporating an existing technique). The proposed method seems effectitive in fine-tuning pre-trained models too, and works for different CNNs including the compact MobileNet. I think this paper will attract profound interests from the NeurIPS community and generate a nice impact. I would like to vote for an acceptance. ------------------------------------------------------ The main limitation of E2Train is currently its loss of accuracy: 0.1% top-5 looks nice; but 2% top-1 acc loss on both CIFAR datasets raise some red flag, even in exchange of 80% - 90% energy savings. I encourage the authors to discuss in rebuttal: Why the performance loss (seems the source is mainly PSF)? Can the accuracy be improved by simple tweaks? If not, what remedies are possible? etc. Both experimental validations and verbal arguments are welcome. The work immediately reminds of the ACl paper "Energy and Policy Considerations for Deep Learning in NLP". That one presented a first step toward raising awareness and quantifying deep learning’s potential CO2 impact. Ever larger models are bound to gobble up energy saved by more efficient architectures and specialized chips. Then, in addition to resource-constrained scenarios, the authors (to their good) might also discuss their algorithms through the lens of reducing carbon footprint of training. Following that thought, it is meaningful to examine whether their algorithms can extend/scale up to even huger models, different hardwares (like TPU), or distributed settings. Besides, I understand this work acts like a pilot study, and therefore standard classification experiments are probably the easiest to benchmark. However, it still would be nice to try at least one adaptation experiment from pre-trained models. Lastly, The paper layout is a bit too compact. Particularly, figures need re-work. For example, Figures 3 and 4 take some time to understand. The three arrows in Fig 5 are confusing.

Reviewer 2



Update after authors' feedback Thanks for the clarifying answers. The rebuttal is clear and addresses many points, but not all of them. It looks like new results will be included, which is nice, but I still think the paper is quite dense and could have been several clearer and lighter submissions. I would have liked more confidence intervals on the results / plots since the improvement or degradation is quite small and could be attributed to randomness. The argument in the rebuttal for the extension of SLU to other architecture is not very convincing and would again still be limited to a subset of models. Also the angle of the paper (on-device training) is not quite aligned with the experiments carried out (I'd have expected continuous training, starting from a pre-trained model). Yet low-energy training is indeed an interesting topic that should be addressed before we burn the planet with deep neural network trainings. For that reason mainly, I'll raise my score to 7, but emphasis that the limitation of SLU to ResNets should be extremely clear in the final version since the title suggest that the method could work with any architecture. ------------------------------------------------------------------------ Overall, this paper is well-written and pretty nice to read. I quite liked the effort of constantly linking, comparing and contrasting the ideas to previous results published in the literature. The topic is a very interesting one. Training or adapting models on-device is an important subject to address nowadays, and the privacy-preserving aspect is quite relevant. Yet, it is hard to believe in that setup that neural networks would be trained from scratch. While the authors point out in the paper, and especially in the conclusion, that the explored scenario is not a realistic one, it would have been interesting to see one or two experiments on adapting or fine-tuning a neural network on-device, to illustrate a real use-case, and compare it for example to training only the last layer, which could be a reasonable baseline. In the explored scenario, the training time is never discussed, and would also be a crucial parameter for the applicability of the proposed methods. That said, as I pointed out earlier, the topic of training of neural networks has raised awareness lately in the media of a quite important issue regarding the energy consumption. This paper could also have been a good contribution, even not limited to the topic of on edge training. Apart from that, I have some reservations. For it to be considered a definite NeuriPS paper, I would have expected either a strong theoretical foundation, a more in-depth analysis or a truly ground-breaking contribution. I am aware that with the page limit there was little room for more and this paper could easily be split into several ones. More specifically: On the stochastic mini-batch dropping: - the results are surprising and to some extent interesting. I'm not aware of the literature on the topic so my comments might be irrelevant, but I have a hard time understanding why the models should be better with that method than with the standard way and the appropriate shuffling and the same amount of data. - regarding the randomness, it looks like results are reported from single runs of experiments. Using for example different random seeds and presenting the results averaged over several runs would have made the results way more solid. Confidence intervals would have been appreciated too. - I did not find the argument at the end of the paragraph on SMD very clear or convincing. - Regarding the experiments, we don't really see the impact of the learning rate schedule and/or the mini-batch size. That would also have been an interesting analysis to provide On the selective layer update: - the author mention that they run experiments with ResNets at several places, but throughout the paper they write about CNNs making it look like all the methods are applicable to all convolutional architectures. Yet I don't see how this method is possible without the skip connections in ResNet, and as such, it should be more explicit that it is only applicable to that family of architectures. - an analysis of what is learned by the auxiliary RNN, predicting which layers to update would have been extremely valuable here too. On the predictive sign gradient: - the experimental part on this aspect is very short. It would have been interesting to see the impact of the chosen precision for the gradients, of the adaptive threshold and for example the proportion of times the full precision gradient is computed (cf. eq. 2), and how it evolves during training. - a bit more details about E1 and E2 in eq.3 would help to read the part on the prediction guarantee (even though those quantities are defined in the appendices) Minor comments: - end of p.7: typo "cn" -> "can" - the resolution of figures and tables should be improved - it might not be that relevant to talk as much about the improvement/degradation of accuracy, especially if the experiments have undergone a single run. The improvements are actually quite small, in the same order of magnitude as what is reported as a negligible accuracy loss on p.7.

Reviewer 3



As most of the concerns have been addressed by the authors' feedback, I upgrade my score. But I still think the authors should continuously improve this paper, at least integrate the claimed new theory and experiment results in their feedback. ############################################################# This paper proposed a novel energy-efficient training framework for CNN that combine three different techniques, namely stochastic mini-batch dropping (SMD), input-dependent selective layer update (SLU) and predictive sign gradient descent (PSG). The authors show the effectiveness of their techniques with several ablation studies as well as real energy measurement on FPGA boards. Results show that this framework can save lots of energy during training with marginal accuracy drops (e.g. when applied to ResNet-74 on CIFAR10, authors get 2% accuracy drop with <10% energy and 1.2% accuracy drop with <40% energy). I don’t find the link to the downloadable source code in the paper and supplementary material as the authors claimed in the reproducibility checklist. Originality: The three new techniques the authors proposed are novel and well prepared with other existing methods. Quality: I think the methods proposed in this paper are rather heuristic with limited experiments support the authors’ claim. SMD: The authors claimed that with the same energy consumed, SMD incur minimal accuracy loss and sometimes even increase the accuracy. However, the authors didn’t show enough experiment results in Section 4. They only showed the top-1 accuracy of CIFAR-10 using ResNet-74, which is not sufficient to support a general result on the whole class of CNN. Moreover, the authors didn’t provide the error bar of the accuracy. I think this conclusion need far more test. If SMD is not consistently better than standard mini-batch training when consumes equal energy, why we need to use it? Some theoretical analysis for SMD maybe better support this conclusion. Moreover, I don’t know the meaning of Figure 3(b). What do the authors want to show? I think beat the accuracy of a larger learning rate standard mini-batch training model in single experiment is not convincing. I think the authors should make this part of experiments solid. SLU: This method is interesting, but a little counter-intuitive. I’m not sure if this technique is suitable for the general CNN architecture without residual connection, at least SkipNet and stochastic depth ResNet is work for only residual network. The authors should make it clear. Moreover, the effect of adding this kinds of regularization is not clear. I can hardly imagine the training dynamics of this method, though maybe useful in practice. In the experiment part, the authors didn’t mention the benchmark model and the dataset. I think it’s still ResNet-74 on CIFAR-10? Still, there’s no error bar, no enough results on other kinds of CNN architecture, and I think SLU can have more interesting experiment result, like the implicit regularization result in the stochastic depth ResNet paper, though this paper focuses on energy-efficient training. Will the adjustment of p_L hurt the performance of stochastic depth ResNet? PSG: I’m a little confused on the description of PSG. The authors argues that SignSGD need to compute the full-precision of gradient before taking the signs, but I think in PSG, we still need to calculate the full-precision gradient g_w as in Eqn. (2)? And what’s the PSG’s predictor’’ in Section 3.3? Throughout the whole paper I don’t find any explanation, I think it’s just the matrix product of x and g_y? Then the whole procedure is predicting the sign of gradient with low-precision input x and g_y first, if cannot determined, then use the full-precision input? As the undetermined elements can be disordered in the whole gradient tensor, and calculate these elements with high-precision vector product can be much slower than matrix product, will this save the computation time (though may save energy, I'm not the expert on hardware)? I think the author should estimate the ratio of undetermined elements, and make this part clear. Authors proved that this procedure have bounded error rate, which is great. Experiments for PSG are also limited to fixed base model ResNet-74. Overall, I think the author proposed strong result for energy-efficient training, however I’m not sure if this result can generalize to models other than ResNet-74 and ResNet-110. The techniques proposed by authors are interesting, but the experiments cannot support their claim well. Clarity: Some part of the paper are not written well. For example, the PSG part I mentioned above seems confusing. The description of proposed methods is too heuristic, and I can hardly understand the mechanism behind those methods. Moreover, I think the experiment section is not well-organized. The authors should pay some attention on making their results solid and inspiring, not restricted to the accuracy and energy consumed on the two ResNet model. I know the paper's length is limited, but you can make some discussion and show extra results in the Appendix. Significance: This paper provide a good empirical results for energy-efficient training.

[Author Response · NeurIPS 2019]

We thank all three reviewers (**R#1, R#3, R#4**) for appreciating our novelty, strong results, and broad impact to the
community, and promise to release all codes upon acceptance and to improve writing clarity. Below, we first respond to
general $E^2$-Train questions, and then specific questions on **SMD**, **SLU** and **PSG**, being grouped between lines.

---

**Reducing accuracy loss? (R#1).** We continued striving to reduce the accuracy loss after NeurIPS submission, and
found a stochastic weight averaging (SWA) technique ("SWALP" paper in ICML'19) to be helpful. As requested by R#1,
we report the improved new result : after applying SWA to $E^2$-Train on ResNet 74, we obtain top-1 93.01% on CIFAR-10
(0.56% loss) and top-1 71.63% on CIFAR-100 (no accuracy loss) with 83.40% and 81.27% energy saving, respectively.
**Experiments on adapting a pre-trained model (R#1, R#3).** We performed a proof-of-concept experiment for CNN
fine-tuning by splitting CIFAR-10 training set into half, where each class was i.i.d. split evenly. We first pre-train
ResNet-74 on the first half, then fine-tune it on the second half. During fine-tuning, we compared two energy-efficient
options: (1) fine-tuning only the last FC layer using standard training; (2) fine-tuning all layers using $E^2$-Train. With all
hyperparameters being tuned to best efforts, the two fine-tuning methods improve over the pre-trained model top-1
accuracy by [0.30%, 1.37%] respectively, while (2) saves 61.58% more energy (FPGA-measured) than (1). That shows
that $E^2$-Train is the preferred option: higher accuracy, and saving much more energy. We will report it in camera-ready.
**On different CNNs (R#3, R#4).** $E^2$-Train is effective in even compact networks, e.g., MobileNetV2: 92.06% / 71.61%
top-1 accuracy on CIFAR-10/-100 (baseline: 92.47% / 71.91%), with 87.73% / 88.17% energy savings, respectively.
**Training time (R#3).** $E^2$-Train takes roughly the same epoch numbers to converge (line 324). Similar to energy, the
per-epoch time is also reduced, thanks to our three-level savings. We will report detailed measurements in camera-ready.

---

**Why SMD works, by theory? (R#3, R#4).** We are happy to confirm that SMD is not simply a heuristic - **we recently**
**proved** that SMD can outperform SGD at certain range of mini-batch sampling ratios and epoch numbers. The proof
is inspired by *"Random Shuffling Beats SGD after Finite Epochs"*, ICML'19, and the core idea relies on the finite
population correction technique. We plan to report our theoretical findings and proof in another upcoming submission.
**Is SMD stable/reproducible? (R#3, R#4).** We repeated training ResNet74 on CIFAR-10 using SMD for 10 times (all
end up saving $\frac{1}{3}$ energy), with different random initializations. The accuracy std. is only 0.132%, showing high stability.
**Is SMD generalizabe to more models? (R#4).** For ResNet74 on CIFAR-100, [SMD, SMB] with $\frac{1}{3}$ energy saving
achieve top-1 [71.37%, 71.11%], respectively; and for ResNet-110 on CIFAR-10, [SMD, SMB] with $\frac{1}{3}$ saving achieve
[93.05%, 92.75%], respectively. Various of such experiment results confirm that SMD is consistently better than SMB.
**Clarifying Fig. 3b (R#4).** We tried to find more baselines for solid evaluation of SMD. As SMD outperformed SMB
with standard learning rates (lr), we conjectured increasing lr might accelerate SMB's convergence (reducing epochs),
thus making another strong baseline. We also repeated SMD/SMB experiments and found their gap to persist (>0.2%).

---

**SLU on non-residual CNNs (R#3, R#4).** We thank both reviewers to point out. Indeed, our current experiments are
based on CNNs with skip connections, partially because they dominate in SOTA CNNs. We will make it clear in
camera-ready. Furthermore, we conjecture SLU can be extended to non-residual CNNs by "appending" skip connections
with gates to plain backbones (i.e., creating "ResNet" versions for their training). We leave it for future work.
**Error bar of SLU (R#4).** 20 trials of SLU experiments to ResNet38 on CIFAR-10 show that, with 95% confidence
level, the confidence interval for the mean of the top-1 accuracy and the energy saving are [92.47%, 92.58%] (baseline:
92.50%) and [39.55%, 40.52%], respectively, verifying SLU's trustworthy effectiveness.
**What auxiliary RNNs learn(R#3)/ SLU training dynamics(R#4).** After training, visualizations show RNNs learned
(i) the class-wise discriminative selectivity of layers (ResBlocks); and (ii) input hardness-aware routing, consistent with
SkipNet's observations. During training, we observe that the auxiliary RNNs converge much faster (<20 epochs) than
the main backbone part, showing "layer selectivity" can be identified at early training stage. Interestingly, this coincides
with observations of "critical learning periods", ICLR'19. We will include the visualizations/analysis into camera-ready.
**Adjusting $p_L$ (R#4).** $p_L$ controls SD's drop ratio, and we always tune it to make SD's drop ratio the same as SLU (line
291). Similar to SLU, dropping more layers decreases SD's performance, in exchange for more energy saving.

---

**Clarifying PSG description (R#4).** PSG is motivated by two facts: (i) reducing precision is very effective (expo-
nentially) for reducing hardware energy cost; and (ii) MSB parts contribute exponentially larger than LSB parts for
calculating outputs. Specifically, while SGD uses $x$ and $g_y$ to calculate $g_w$, PSG uses $x^{\mathrm{msb}}$ and $g_y^{\mathrm{msb}}$ to calculate $g_w^{\mathrm{msb}}$
for predicting the sign of $g_w$ when $|g_w|^{\mathrm{msb}} \geq \tau$ (this happens with high probability thanks to fact (ii), e.g., > 60% in
ResNet74 on CIFAR10). Otherwise (i.e., $|g_w|^{\mathrm{msb}} < \tau$), PSG will proceed to finish the remaining residual computation
of calculating $g_w$ and its sign. In hardware, the computation to obtain $g_w^{\mathrm{msb}}$ is embedded within that of $g_w$. Therefore,
the PSG's predictors (i.e., calculating $g_w^{\mathrm{msb}}$ using $x^{\mathrm{msb}}$ and $g_y^{\mathrm{msb}}$) do not incur any energy overhead.
**More requested details on PSG (R#3).** We use default $\beta = 0.05$ to control MSBs, while the effectiveness of PSG is
observed to be insensitive to the choice of $\beta$ when it is in the range of [0.05, 0.1]. If $\beta$ is too small, it will result in too
frequent coarse gradients and might hurt convergence. The ratio of using merely $g^{\mathrm{msb}}$ typically remains at least 60% in
the training process. We will include these in camera-ready, along with detailed explanation of $E_1$ and $E_2$ in (3).

[Meta-Review · NeurIPS 2019]

This paper addresses an important topic of energy-efficient training of CNNs. They investigate this in three different levels and report promising results on hardware testbed. The reviewers did raise a number of critical questions, especially on theoretical understanding of *why* the proposed bag of tricks work and if good results can be easily generalized and reproduced. The rebuttal provided informative and convincing to the reviewers comments and promise to an number of improvement. We recommend accept the paper, but strongly encourage the authors to carefully address the reviewers' concerns. We also strongly encourage the authors to make their code publicly available; we think this is especially critical given that this work is mainly empirically and the results can not be verified or reproduced without code available.